

# The effect of social information from live demonstrators compared to video playback on blue tit foraging decisions

Liisa Hämäläinen[1], Hannah M. Rowland[1,2,3], Johanna Mappes[4] and Rose Thorogood[1,5,6]

[1] Department of Zoology, University of Cambridge, Cambridge, UK
[2] Institute of Zoology, Zoological Society of London, London, UK
[3] Max Planck Institute for Chemical Ecology, Jena, Germany
[4] Department of Biological and Environmental Sciences, University of Jyväskylä, Jyväskylä, Finland
[5] HiLIFE Helsinki Institute of Life Science, University of Helsinki, Helsinki, Finland
[6] Research Programme in Organismal & Evolutionary Biology, Faculty of Biological and Environmental Sciences, University of Helsinki, Helsinki, Finland

Corresponding author
Liisa Hämäläinen, llh35@cam.ac.uk

## ABSTRACT

Video playback provides a promising method to study social interactions, and the number of video playback experiments has been growing in recent years. Using videos has advantages over live individuals as it increases the repeatability of demonstrations, and enables researchers to manipulate the features of the presented stimulus. How observers respond to video playback might, however, differ among species, and the efficacy of video playback should be validated by investigating if individuals' responses to videos are comparable to their responses to live demonstrators. Here, we use a novel foraging task to compare blue tits' (*Cyanistes caeruleus*) responses to social information from a live conspecific vs video playback. Birds first received social information about the location of food, and were then presented with a three-choice foraging task where they could search for food from locations marked with different symbols (cross, square, plain white). Two control groups saw only a foraging tray with similar symbols but no information about the location of food. We predicted that socially educated birds would prefer the same location where a demonstrator had foraged, but we found no evidence that birds copied a demonstrator's choice, regardless of how social information was presented. Social information, however, had an influence on blue tits' foraging choices, as socially educated birds seemed to form a stronger preference for a square symbol (against two other options, cross and plain white) than the control birds. Our results suggest that blue tits respond to video playback of a conspecific similarly as to a live bird, but how they use this social information in their foraging decisions, remains unclear.

## INTRODUCTION

The number of studies investigating social information use in animals has been expanding during the last few decades, and it is now well documented that many species use social

information in their decision-making (*Galef & Laland, 2005*). Acquiring social information can be beneficial in many different contexts. Animals can, for example, use social information in their foraging decisions, mate choice, breeding habitat selection, or when avoiding predators (*Danchin et al., 2004*). Social transmission is taxonomically widespread, with evidence of social information use found in birds (*Aplin, 2019*), mammals (*Whiten, 2000*), fish (*Brown & Laland, 2003*), reptiles (*Noble, Byrne & Whiting, 2014*; *Kis, Huber & Wilkinson, 2015*) and insects (*Dawson & Chittka, 2012*; *Baracchi et al., 2018*). Social information is predicted to benefit individuals by reducing the costs of personal learning (*Laland, 2004*; *Kendal et al., 2005, 2018*). When foraging, for example, individuals can gather social information about the location of food sources or food palatability, and learn novel foraging skills (reviewed in *Galef & Giraldeau, 2001*), which could increase their foraging efficiency.

As the number of social learning studies has grown, also the number of techniques to study social interactions has increased. A common method is to use artificial stimuli that enables researchers to control and standardise what information is presented (*D'Eath, 1998*; *Woo & Rieucau, 2011*). Artificial stimuli have been used for a long time in animal behaviour research, starting from simple dummies and leading up to robotic animals. Cardboard models were first used by *Tinbergen & Perdeck (1950)* to investigate the importance of various stimulus characteristics on the begging response of herring gull chicks. Subsequently, simple models have been used in many experiments, including studies investigating mate choice (*Halnes & Gould, 1994*; *Höglund et al., 1995*), or individuals' responses to predators (*Powell, 1974*; *Petersson & Järvi, 2006*) and brood parasites (*Thorogood & Davies, 2016*). Over the recent years, new technology has enabled researchers to use also more sophisticated techniques, such as robotic animals (*Taylor et al., 2008*; *Krause, Winfield & Deneubourg, 2011*). For example, male satin bowerbirds were found to adjust their displays in response to signals from robotic females (*Patricelli et al., 2002*), and wild grey squirrels were shown to respond to a robotic model of a conspecific displaying alarm behaviour (*Partan, Larco & Owens, 2009*).

Another promising technique to study social interactions is video playback. Videos can be easily edited and manipulated, allowing researchers to alter the stimulus features that are presented to observers and reduce the variation among presentations (*D'Eath, 1998*). Video presentations can be used to study animals' responses to simple animations, such as point-light displays, and domestic chicks (*Gallus gallus domesticus*) have been demonstrated to prefer biological motion patterns when exposed to these displays (*Vallortigara, Regolin & Marconato, 2005*; *Vallortigara & Regolin, 2006*). Furthermore, with technological advances it is now possible to create realistic computer-generated animations of animal models to study social interactions (*Woo & Rieucau, 2011*). However, a more common method in behavioural studies is to record a video of a live animal and video playback has now been used successfully in many bird species (*Adret, 1997*; *Ikebuchi & Okanoya, 1999*; *Ophir & Galef, 2003*; *Bird & Emery, 2008*; *Rieucau & Giraldeau, 2009*; *Guillette & Healy, 2017*; *Thorogood, Kokko & Mappes, 2018*; *Carouso-Peck & Goldstein, 2019*; *Smit & Van Oers, 2019*), as well as across a range of other taxa, including mammals (*Hopper, Lambeth & Schapiro, 2012*; *Gunhold, Whiten & Bugnyar,*

2014), fish (*Rowland et al., 1995*; *Trainor & Basolo, 2000*), reptiles (*Clark, Macedonia & Rosenthal, 1997*; *Ord et al., 2002*) and spiders (*Clark & Uetz, 1992*). Video playback does, however, have limitations such as the lack of depth cues, the lack of interaction between an observer and an individual on the video, and differences between animal and human visual systems (*D'Eath, 1998*; *Zeil, 2000*; *Ware, Saunders & Troje, 2015*). Birds, for example, have higher critical flicker-fusion frequencies (>100 Hz) than humans (60 Hz) and they might therefore perceive the video image as flickering, instead of continuous motion (*D'Eath, 1998*; *Bird & Emery, 2008*). However, this degree of visual resolution often occurs when light stimuli are very bright (e.g. 1,500 cd/m$^2$ in blue tits (*Boström et al., 2016*)) and beyond the normal brightness of most video screens. Furthermore, the use of liquid crystal display (LCD) monitors instead of older cathode ray tube displays can help to overcome the problem of flicker, and especially a flickerless thin film transistor LCD has provided a good method to present videos to birds (*Ikebuchi & Okanoya, 1999*). Another important aspect to take into account is image presentation rate (IPR) which influences how realistic the motion on the video appears (*Ware, Saunders & Troje, 2015*). *Ware, Saunders & Troje (2015)* demonstrated that pigeons (*Columbia livia*) responded to videos of a conspecific more strongly when IPR was 60 frames per second, compared to lower presentation rates (15 or 30 frames/s), and the authors therefore suggest researchers to use the highest frame rate available when using video playback.

Although videos have been used successfully in many studies, video playback does not always generate the same responses in observers when compared to studies using live demonstrators (see *Schlupp, 2000*). For example, a recent study with California scrub-jays (*Aphelocoma californica*) found that observing a video of a conspecific eavesdropping on a caching event did not influence focal individuals' caching and re-caching behaviour, in contrast to previous studies with a live conspecific (*Brecht et al., 2018*). The strength of the responses to video and live demonstrations may also differ even when observers are found to respond to videos. Zebra finch (*Taenopygia guttata*) males, for example, copy the nest material choice from a video demonstrator but this preference is stronger when birds observe a live demonstrator (*Guillette & Healy, 2019*). Most of these studies, however, have compared individuals' responses to video playback to previous experiments with live demonstrators, and therefore have not accounted for possible differences in test conditions, such as individual differences among the demonstrators. Here our aim was to compare these two methods in one study by investigating whether blue tits' response to the same demonstrator differs between video and live presentation.

The applicability of video playback in studies with blue tits is so far unclear. We found recently that blue tits' behaviour changed when they were presented with video playback of a conspecific, but social information from videos did not influence their foraging decisions in a later foraging task (*Hämäläinen et al., 2017*). In contrast, great tits (*Parus major*) have been demonstrated to respond to videos of a conspecific (*Snijders, Naguib & Van Oers, 2017*), and use social information from videos in their foraging decisions (*Thorogood, Kokko & Mappes, 2018*; *Hämäläinen et al., 2019*; *Smit & Van Oers, 2019*), suggesting that video playback can be used successfully in other parid tit species. It is, however, possible that even closely related species differ in their response to video stimuli.
For example, *Roberts, Gumm & Mendelson (2017)* tested the efficacy of video playback in two species of darters, *Etheostoma barrenense* and *Etheostoma zonale*, and found that despite the same experimental set-up and close relatedness of the species, only *E. zonale* females' responses to video playback of conspecific males were comparable to live males, whereas *E. barrenense* females showed a preference only for live males. Similarly, blue tits might respond to videos differently than great tits. Alternatively, our previous result of blue tits not copying a demonstrator (*Hämäläinen et al., 2017*) might be because blue tits were simply not using acquired social information, regardless of how it was presented. Indeed, studies using live demonstrators have found that only about half of the tested blue tits learn a novel foraging task socially (*Sasvári, 1979*, *1985*; *Aplin, Sheldon & Morand-Ferron, 2013*), compared to great tits that are more likely to solve the task after observing others (*Sasvári, 1979*, *1985*). To disentangle the effect of video playback and blue tits' tendency to use social information, we designed an experiment where we investigated whether birds were more likely to use social information from a live demonstrator, compared to a video presentation.

In this experiment, we presented blue tits with a three-choice foraging task: an ice cube tray with three wells covered and marked with different symbols (cross, square and plain white). One group of the birds received social information about the location of food from a live conspecific, whereas another group saw a video playback of a conspecific demonstrator. In addition, we had two control groups that only saw a foraging tray (live/video presentation) but no information about the location of food. We predicted that the birds in the control group would not have a preference for any of the symbols and would choose each of them equally often. Socially educated birds were predicted to choose the same symbol and location where they had observed a demonstrator foraging. We predicted that blue tits would copy a demonstrator's choice equally often regardless of how social information was presented (live/video demonstrator). However, finding that blue tits were less likely to copy a demonstrator's choice from videos would indicate that video playback might not be a suitable method for social learning studies in the species. Finally, we predicted that birds that received social information would start the foraging task faster than control birds (*Hämäläinen et al., 2017*; *Thorogood, Kokko & Mappes, 2018*).

## METHODS

### Birds

The experiment was conducted at Konnevesi Research Station in Central Finland during January and February 2017. We tested social information use in 40 juvenile blue tits. In addition, five adult birds were used as demonstrators. Birds were caught from the feeding site and housed in individual plywood cages (80 cm (h) × 65 cm (w) × 50 (d) cm) with a daily light period of 12.5 h, and free access to food (sunflower seeds, tallow and peanuts) and fresh water. Before and during the experiment food was restricted to make sure that birds were motivated to forage. Birds were kept in captivity for approximately 1 week and then released back at the capture site. Before this, each bird was weighed and ringed for identification purposes. The work was carried out with permission from the
Central Finland Centre for Economic Development, Transport and Environment and licence from the National Animal Experiment Board (ESAVI/9114/ 04.10.07/2014) and the Central Finland Regional Environmental Centre (VARELY/294/2015). Birds were treated following the ASAB guidelines for the treatment of animals in behavioural research and teaching (2012).

## Foraging task and pre-training

We investigated whether blue tits used social information about the location of food by presenting them with a three-choice foraging task where they had to find mealworms from a white plastic ice cube tray (modifying a protocol used in *Hodgson & Healy, 2005*). The tray had 21 wells in three rows and we covered three of these (in the middle row) with a piece of white paper that had either (i) a black cross symbol, (ii) a black square symbol, or (iii) no symbol (plain white) printed on top (Fig. 1A). The same symbols were attached in front of the foraging tray to increase their visibility to the observers during demonstration. In the experiment birds had to lift up the paper covers to find a food reward and we investigated whether social information influenced their first choice.

Before the experiment, we trained birds in their home cages to forage from an ice cube tray. The training was done step-wise by first offering birds a tray with four of the wells (randomly chosen) containing a mealworm. After birds had eaten these, we next presented them with a tray with four wells partly covered (again, randomly chosen), so that the mealworms were still visible. During the training, we covered the wells with brown paper to prevent birds associating the reward with white colour that was used in the social learning experiment. In the next step birds received a tray where four wells were covered with brown paper, so that the mealworms were completely hidden. After birds had completed these steps (i.e. found and consumed all mealworms), we finally presented them with a tray with seven wells covered but only four of them containing a mealworm. This was done to increase individuals' uncertainty about a food reward in the wells, so that they would be more likely to use social information in the experiment. Training was completed once individuals had found and consumed all mealworms. All birds finished the training in 1 day.

## Demonstrators

We used five individuals (all adults, i.e. >1 year old) as demonstrators in the experiment. Each individual was used twice in the live demonstration and also filmed for the video playback that was presented to two observers (i.e. each individual was demonstrator for four observers). Demonstrators were first trained to forage from an ice cube tray in their home cages, following a similar step-wise protocol that we used with observers (see above). However, instead of covering the wells with brown paper, we presented demonstrators with a similar tray that we used in the experiment, with three wells covered with different symbols (cross, square, plain white; Fig. 1A). The food reward was placed only under one of the symbols (cross or square) whereas the other wells were always empty. Demonstrators therefore learned to associate a food reward with one of the symbols and

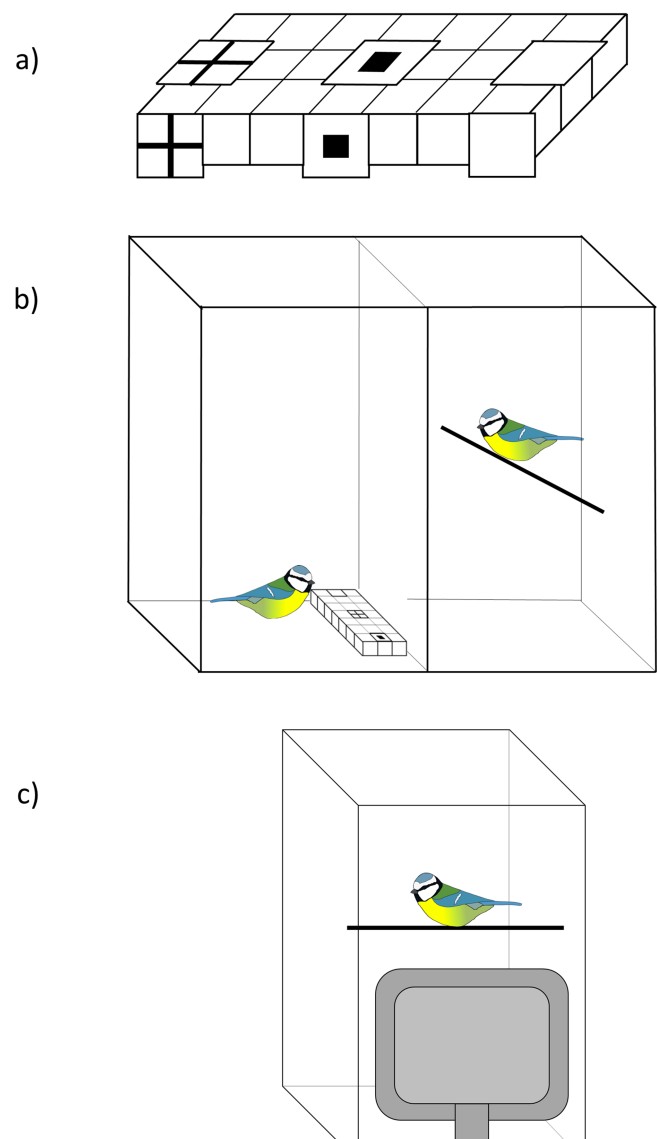

**Figure 1 The experimental set-up.** (A) An example of the ice cube tray that was presented to birds. The tray had 21 wells and three of them (left, middle and right well in the middle row) were covered with a piece of white paper that had either a black cross or a square printed on top, or no symbols (plain white). The same symbols were attached in front of the tray to increase their visibility to observers. The order of the symbols was randomised among birds. (B) The set-up of the live demonstration. The demonstrator (left) and the observer (right) were in individual cages that were separated by plexiglass, so that birds could see each other. In the control treatment the birds saw only the tray. (C) The set-up of the video playback. A computer monitor was placed against a plexiglass front wall of the test cage. Birds were then presented a video of a demonstrator or a control video of the tray. Blue tit illustration credit: Victoria Franks.

searched for food from that location during the demonstrations. We trained two of the demonstrators to associate a food reward with a cross symbol, and two with a square symbol. To ensure that the number of demonstrations for each symbol was balanced, the last of the five demonstrators was trained first with a square and then with a cross.

For the video playback, we filmed each demonstrator performing the foraging task (i.e. finding a mealworm by lifting up the paper cover) through the plexiglass wall of the test cage (a 66 (h) × 50 (w) × 50 (d) cm sized plywood cage with the plexiglass front wall) using an HD camcorder Canon Legria HF R66 (with 50 frames/s progressive recording mode). Three mealworms were hidden in the well (with either a cross or a square symbol), and birds were filmed finding and eating all of them, so the demonstration was repeated three times. We then edited these videos (using Windows Movie Maker), so that they were all 150 s long (see a video clip in Supplemental Material). We also filmed a 5-min long video of a demonstrator in the cage without a tray, which was presented to observers before the foraging task demonstration. Finally, we filmed control videos that contained a tray only (with different symbols) but no bird (150 s). We filmed six different control videos with all possible symbol orders on the tray to ensure that the location on the tray would not influence our results.

## Experimental protocol

In the experiment observers were randomly allocated to four treatments ($n = 10$ in each): (i) social information from a live demonstrator, (ii) social information from video playback, (iii) live control (the feeding tray only), (iv) video playback control (video of the feeding tray only). In all treatments, birds were first allowed to habituate to the test cage for 2 h. During this time, we repeated the foraging task training one more time by presenting birds with an ice cube tray with seven wells covered with brown paper and four of these containing a mealworm. After this food was restricted for 1 h which is a moderate level of deprivation for blue tits and increases their motivation to search for food during the experiment.

The live demonstration was conducted in a plywood cage that was divided into two individual compartments (each 66 (h) × 50 (w) × 50 (d) cm) that were separated by a plexiglass wall (Fig. 1B). An individual that was tested was placed on one side of the wall, and a demonstrator bird (or a tray only for the control group) on the other side. Outside the experiment, the plexiglass was covered (with a cardboard sheet), so that the birds could not see each other, and the cover was removed only for the duration of the demonstration. The front wall of each compartment was similarly made of plexiglass, so that we could observe the birds during the experiment. The demonstrator was placed in the test cage 2 h before the test (with plexiglass between the two cage compartments covered). Demonstrators were then given one more training session with the symbols to ensure that they were foraging in the test cage, and that they were choosing the right symbol (the symbol they had been trained to associate with a reward). After this, demonstrators were food-deprived for 1 h, so that they were motivated to forage during the demonstration. We then removed the cover of the plexiglass between the observer and the demonstrator, and let the birds to habituate to this new situation for 5 min before presenting the foraging tray to the demonstrator. The tray had three wells covered and one of them (the well with either a cross or a square symbol) contained three mealworms. The order of the symbols was randomised across presentations. We waited until the

demonstrator found and ate all three mealworms which took on average 230 s (range = 154–492 s).

Once the demonstration was finished (i.e. the demonstrator had consumed all three mealworms), we covered again the plexiglass between the cages, so that the birds could not see each other. We then presented observers with a foraging tray with the same three symbols. The order of the symbols in the presented tray was the same as in the demonstration, so that observers could use both symbol and spatial cues about the location of the food reward. This time all the wells were empty to make sure that birds could not get any additional cues about food. We recorded observers' first choice to search for food (i.e. the well where they first lifted up the cover) and the test was finished after this. To investigate whether social information influenced the birds' latency to start the task, we also recorded the elapsed time (s) before the choice. The live control treatment was conducted in a similar way but instead of seeing a demonstrator, birds saw only the tray in an empty cage for 150 s.

When birds received information from videos, the experiment was conducted in a 66 (h) × 50 (w) × 50 (d) cm sized plywood cage with the front wall made of plexiglass. We presented birds videos by placing an LCD monitor (Dell E198FPF, 19", resolution 1,280 × 1,024, 75 Hz refresh rate, 300 cd/m$^2$) against the plexiglass (Fig. 1C), following previously validated methods (*Hämäläinen et al., 2017*; *Thorogood, Kokko & Mappes, 2018*; *Hämäläinen et al., 2019*). The size of the demonstrator on the screen was smaller than the size of the live bird (approximately 70% of the real size). How birds perceive the demonstrator's size is, however, difficult to estimate because of depth cues (*Zeil, 2000*) and differences in viewing distance, depending on an observer's position in the cage. Nevertheless, previous studies have demonstrated that great tits use social information from the videos with a similar sized demonstrator (*Thorogood, Kokko & Mappes, 2018*; *Hämäläinen et al., 2019*). Birds were first let to habituate to the monitor for 15 min before starting the video. Birds that received social information were then presented a 5-min video of a demonstrator in the cage without the foraging tray, so that the protocol was similar to the live demonstration treatment where birds could observe each other for 5 min before the demonstration. Birds were then presented with a 150 s long video of a demonstrator finding and consuming three mealworms under one of the symbols. Birds in the control group saw a video of the feeding tray only (150 s). After this, the computer monitor was removed and we presented birds with the foraging task, following the same protocol as in live demonstration. Again, the order of the symbols was the same as on the videos, and we recorded birds' first choice and the time before they started the task.

## Statistical analyses

We first investigated whether birds had an overall preference towards any of the symbols using a binomial test (compared to equal probability of choosing any of the three symbols). We then investigated whether these preferences differed between socially educated and control birds. Because we did not find differences in information use between video and live demonstration treatments (see 'Results'), we combined these treatments and used a *G*-test to compare distributions of the preferences between all socially educated birds

(live and video treatment; $n = 20$) and control birds (live and video treatment; $n = 20$ ). We also used a *G*-test to investigate (i) if birds had a preference for the spatial location on the tray (left/middle/right), i.e. if they chose any of the locations more often than expected by chance (1/3 probability) and to (ii) compare the choices of socially educated birds that saw a demonstrator choosing a square to those seeing a demonstrator choosing a cross (video and live treatments combined). Because birds seemed to prefer a square symbol (see 'Results'), we did this by comparing the likelihoods to choose a square (over alternative options cross/white), i.e. testing if birds chose a square more often after seeing a demonstrator choosing it, compared to seeing a demonstrator choosing a cross. We next used a Fisher's exact test to investigate if birds were more likely to copy a demonstrator's choice when they were (i) presented a live demonstrator, compared to video playback and (ii) when a demonstrator chose a square, compared to a cross. This was done by simply comparing the number of birds whose choice matched that of a demonstrator to those who chose a different symbol. Finally, we tested if social information influenced the latency to start the foraging task using a Cox regression analysis. The time to choose the well (s) was used as a response variable and this was explained by an interaction term between social information treatment (social information/control) and the way information was presented (live/video demonstration). Other explanatory variables in the model included the symbol (cross/square/white) and the tray location (left/middle/right) that the birds chose. To investigate whether birds that matched a demonstrator's choice started the foraging task faster than those that did not, we also conducted the analysis including only socially educated birds (live and video treatment; $n = 20$). The latency to choose was again used as a response variable and this was explained by an interaction term between information type (live/video demonstrator) and whether birds chose a same symbol as a demonstrator or not. All analyses were conducted with the software R.3.3.1 (*R Core Team, 2016*), using *survival* package (*Therneau, 2015*).

## RESULTS

Overall, birds chose the well with a square symbol more often than predicted by chance (binomial test, 25/40, $p < 0.001$). This preference, however, differed between socially educated and control birds (*G*-test, $G = 7.16$, $p = 0.028$; Fig. 2A): individuals that received social information (live and video treatments combined) showed a strong preference towards a square symbol (binomial test, 15/20, $p < 0.001$), whereas this preference was not significant in the control groups (binomial test, 10/20, $p = 0.15$). Against our prediction that socially educated birds would choose the same symbol as a demonstrator, we did not find evidence that a demonstrator's choice (cross/square) influenced an observers' likelihood to choose a square symbol (*G*-test, $G = 0.51$, $p = 0.47$). Instead, socially educated birds seemed to prefer a square, regardless of a demonstrator's choice (Fig. 2A). This did not differ between live and video presentations, i.e. birds were not more likely to copy the choice of a live demonstrator compared to video playback (Fisher's exact test, $p = 1$; Fig. 2B). Because socially educated birds preferred a square symbol, they were found to be more likely to match a demonstrator's choice when a demonstrator chose a square

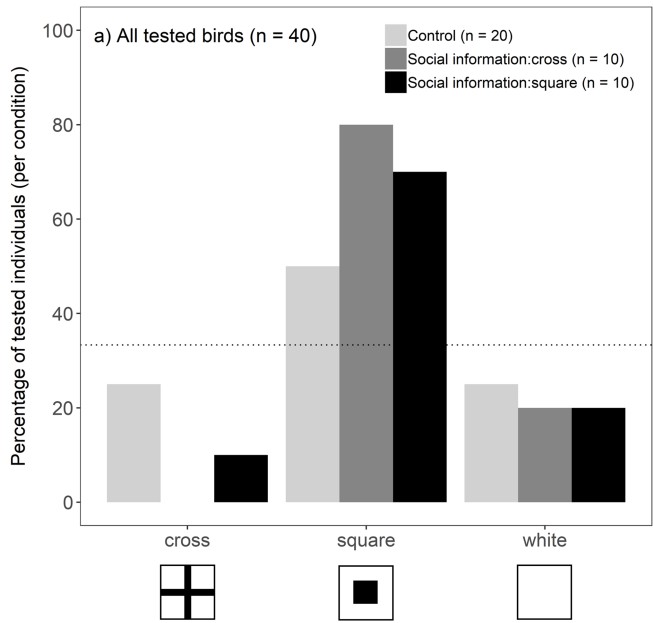

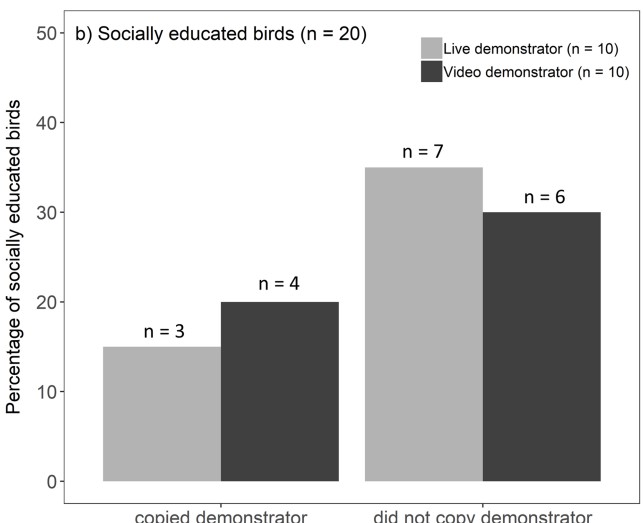

**Figure 2 Birds' foraging choices in the experiment.** (A) The percentage of birds (*n* = 40) choosing each symbol when they were presented (live and video demonstrations combined) (i) a tray only (light grey bars, *n* = 20), (ii) social information of a demonstrator choosing a cross (dark grey bars, *n* = 10), or (iii) social information of a demonstrator choosing a square (black bars, *n* = 10). In the absence of any symbol preference by the birds, each symbol was predicted to be chosen with 1/3 probability. This is represented by the dotted line (33%) and the bars above the line indicate birds' preference towards that symbol. (B) Percentage of socially educated birds (*n* = 20) that copied the demonstrator (i.e. chose the same symbol as a demonstrator vs one of the other two symbols) after seeing a live demonstrator (light grey bars, *n* = 10) or video playback of a demonstrator (dark grey bars, *n* = 10).

symbol compared to a demonstrator choosing a cross (Fisher's exact test, *p* = 0.003). The location on the tray (left/middle/right) did not influence birds' choices (location that birds chose did not differ from that expected by random chance; *G*-test, *G* = 3.62, *p* = 0.16).

Both control and socially educated birds started the foraging task faster after seeing a video demonstration, compared to live demonstration groups (effect of video presentation: coefficient = 1.072 ± 0.420, $Z$ = 2.553, $p$ = 0.01). Birds that chose the right side of the tray also initiated the task faster than birds that chose the left side (effect of location (right): coefficient = 1.086 ± 0.458, $Z$ = 2.372, $p$ = 0.02). Birds tended to choose a square symbol faster than a cross symbol, but this effect was marginal (effect of symbol (square): coefficient = 0.918 ± 0.526, $Z$ = 1.745, $p$ = 0.08). Received social information did not influence how fast birds started to forage (effect of social information: coefficient = −0.210 ± 0.382, $Z$ = −0.549, $p$ = 0.58), regardless of the way the information was presented (social information * type of presentation (video): coefficient = 0.265 ± 0.726, $Z$ = 0.366, $p$ = 0.71), and these non-significant terms were removed from the final model. However, when investigating only socially educated birds, we found that birds that matched a demonstrator's choice started the foraging task more quickly (mean = 81 s, range = 12–253 s) than those that did not (mean = 768 s, range = 35–2,640 s; matching a demonstrator: coefficient = 1.058 ± 0.539, $Z$ = 1.962, $p$ = 0.049). This did not depend on the way information was presented (matching a demonstrator * type of presentation (video): coefficient = −0.635 ± 0.970, $Z$ = −0.655, $p$ = 0.51), and this interaction was excluded from the final model.

## DISCUSSION

In this experiment, we tested whether blue tits were more likely to copy the food choice of a live conspecific, compared to video playback. However, we found that blue tits did not copy a demonstrator's choice of symbol, regardless of how social information was presented. Instead, individuals chose the well with a square symbol more often than other options (Fig. 2A). Because of this preference and the lack of evidence that observers copied a demonstrator's choice, it is difficult to compare the effectiveness of video playback and live demonstration. However, birds' preference for a square symbol was stronger after they received social information, compared to the control groups, and birds whose choice matched that of their demonstrator were quicker to initiate foraging. These responses were consistent across both social information treatments, indicating that even if birds did not often choose the same symbol as a demonstrator, they responded to video and live presentations similarly.

Blue tits might not value social information when the foraging task is relatively simple. Similar to our previous video playback study (Hämäläinen et al., 2017), we did not find evidence that blue tits copied the foraging choice of a conspecific from the video, and neither did they copy the choice of a live demonstrator. Other studies with live demonstrators have similarly failed to find a strong effect of social information in blue tits, showing that only approximately 50% of tested birds learn a novel foraging task socially (Sasvári, 1979, 1985; Aplin, Sheldon & Morand-Ferron, 2013). Social learning seems to also be age- and sex-biased with juveniles (Sasvári, 1985) and especially juvenile females being more likely to learn socially (Aplin, Sheldon & Morand-Ferron, 2013). To increase the chances of detecting social information use, we therefore decided to test only juveniles, but we were not able to determine the sex of the tested individuals. Furthermore, birds
were provided with both visual and spatial cues about the food reward (the location of the symbols in the foraging task mirrored that in the demonstration), so individuals could have used either type of information. Despite this, we failed to find evidence of blue tits copying a demonstrator's foraging choice. However, similar to our previous study (*Hämäläinen et al., 2017*), we found that birds that matched a demonstrator's choice started the foraging task more quickly than birds that chose an alternative symbol, suggesting that social information did influence their behaviour. In addition, birds started the task faster after seeing video playback (either control or social information) compared to seeing live stimuli. This probably results from slight differences between the test conditions (i.e. different test cages). After the live demonstration, we covered the observer's view of the demonstrator's cage by sliding a cardboard sheet between the two cage compartments, and this disturbance might have affected the observers more than simply removing the computer monitor following the video demonstration. Therefore, the test with live stimuli might have been slightly more stressful for the birds which could explain the longer hesitation to start the foraging task.

Despite failing to find evidence that blue tits copied the foraging choice of a demonstrator, social information did have an influence on their foraging choices. In all treatments, birds chose the square symbol more often than other two options (cross or white). However, this preference for squares was even stronger when birds received social information from a live or video demonstrator, regardless of the demonstrator's choice. This indicates that simply seeing a demonstrator foraging from the tray enhanced blue tits' preference towards the square symbol. This result is difficult to explain, but it is possible that birds saw a demonstrator as a competitor, which led them to choose the most visible and preferred prey item. Blue tits were similarly found to prefer squares in another experiment, where birds were allowed to choose between two prey items with cross and square symbols (L Hämäläinen, 2019, unpublished data). A conspicuous square therefore seems to be a more salient cue for blue tits, and contrasting social information about food location did not override this preference. Great tits were recently found to have a high level of self-control ability (*Isaksson, Urhan & Brodin, 2018*), but to our knowledge this has not been tested in blue tits, and it is possible that blue tits were simply too impulsive to inhibit their response to the preferred signal. This initial preference makes our results difficult to interpret, and different symbols might have provided us better evidence of social information use. Interestingly, the preference for square symbols has not been found in great tits (*Lindström et al., 2001*; *Hämäläinen et al., 2019*), and artificial prey with cross and square symbols have been used in many avoidance learning experiments (*Alatalo & Mappes, 1996*; *Lindström et al., 1999, 2001*; *Thorogood, Kokko & Mappes, 2018*). In these experiments squares often represent unpalatable aposematic prey and great tits acquire avoidance to squares faster after receiving social information about their unpalatability (*Thorogood, Kokko & Mappes, 2018*; *Hämäläinen et al., 2019*). Despite the initial preference for squares, blue tits similarly learn to avoid them faster after observing a negative foraging experience of a conspecific (L Hämäläinen, 2019, unpublished data) which shows that blue tits can switch their foraging preferences according to acquired social information. However, our experiment suggests that this is context-dependent, and

blue tits do not change their preferences when they receive positive social information and the foraging task is relatively simple.

Our study highlights the importance of comparing animals' response to real and video stimuli when testing the applicability of video playback (*D'Eath, 1998*). Without the live demonstrator treatment, it would have been difficult to separate the effect of video presentation from blue tits' tendency to use social information. However, because birds were not more likely to copy the choices of live demonstrators, we can now be more confident that our result is not explained only by the lack of response to video playback. Comparing individuals' responses between video and live demonstrations is important even when videos are found to have an effect on observers' behaviour, as these responses could be different compared to live stimuli. The responses to videos might also be context-dependent: zebra finch males showed a stronger preference for the nest material choice of a live conspecific (*Guillette & Healy, 2019*), whereas female zebra finches courted video images of males more actively than live males, possibly because of the lack of reciprocal response from males on the video (*Swaddle, McBride & Malhotra, 2006*). The efficacy of video playback seems to also depend on the features of the video presentation, such as the sound on the video. Zebra finches were shown to copy foraging choices from video playback only when videos did not have sound (*Guillette & Healy, 2017*), whereas the opposite was true in Burmese red junglefowl (*Gallus gallus spadecius*) that used social information only from videos that included sound (*McQuoid & Galef, 1993*). Together, these studies indicate that video playback can be a useful tool in behavioural studies but its applicability might vary among species and different contexts.

## CONCLUSION

The aim of our study was to test the effectiveness of video playback in social learning studies in blue tits by comparing social information use between live and video demonstrations. This comparison proved to be difficult, as we did not find strong evidence of social learning from either live or video demonstrators, indicating that blue tits do not rely on social information in simple foraging tasks. In our experiment the cost to search for food (i.e. lift up the paper cover) was probably low and birds might have ignored social information because personal information was easy to acquire (*Laland, 2004*; *Kendal et al., 2005*). It is also possible that birds would have needed to observe several demonstrations from different individuals before relying on social information. In our experiment individuals received information from one demonstrator only, whereas in nature blue tits form foraging flocks and have opportunities to gather information from both conspecifics and heterospecifics (*Farine et al., 2015*). Individuals are also likely to vary in their tendency to use social information (*Sasvári, 1979*; *Aplin, Sheldon & Morand-Ferron, 2013*) and we might have needed a bigger sample size to detect social learning. Furthermore, instead of using positive social information about the location of food, some observers might have seen the demonstrator as a competitor and therefore avoided the same symbol. Nevertheless, we found that blue tits responded to video playback similarly to a live demonstrator, as both demonstrations enhanced observers' preference towards squares, indicating that videos had the same effect on birds' behaviour as live

demonstrators. However, because of the difficulties to detect social learning in blue tits, the efficacy of videos should be tested in other contexts before making conclusions of its applicability for this species.

## ACKNOWLEDGEMENTS

We thank Helinä Nisu and staff at the Konnevesi Research Station for taking care of the birds and providing facilities for the experiments, Victoria Franks for providing a blue tit illustration and Lucy Aplin for discussions on the methodology. The manuscript was improved by comments from Patrice Adret and the anonymous referee, as well as three anonymous reviewers at Peerage of Science.

### Funding

Liisa Hämäläinen was funded by the Finnish Cultural Foundation and Emil Aaltonen Foundation. Hannah Rowland was supported by a research fellowship from the Institute of Zoology, and is currently supported by the Max Plank Society. Johanna Mappes was supported by the Academy of Finland (#284666 and #320438) and the University of Jyväskylä. Rose Thorogood was supported by an Independent Research Fellowship from the Natural Environment Research Council UK (NE/K00929X/1) and a start-up grant from the Helsinki Institute of Life Science (HiLIFE), University of Helsinki. The funders had no role in study design, data collection and analysis, decision to publish, or preparation of the manuscript.

### Grant Disclosures

The following grant information was disclosed by the authors:
Finnish Cultural Foundation and Emil Aaltonen Foundation.
Institute of Zoology, and the Max Plank Society.
Academy of Finland (#284666 and #320438) and the University of Jyväskylä.
Independent Research Fellowship from the Natural Environment Research Council UK (NE/K00929X/1) and a start-up grant from the Helsinki Institute of Life Science (HiLIFE), University of Helsinki, Helsinki, Finland.

### Competing Interests

The authors declare that they have no competing interests.

### Author Contributions

- Liisa Hämäläinen conceived and designed the experiments, performed the experiments, analysed the data, contributed reagents/materials/analysis tools, prepared figures and/or tables, authored or reviewed drafts of the paper, approved the final draft.
- Hannah M. Rowland conceived and designed the experiments, contributed reagents/materials/analysis tools, authored or reviewed drafts of the paper, approved the final draft.

- Johanna Mappes conceived and designed the experiments, contributed reagents/ materials/analysis tools, authored or reviewed drafts of the paper, approved the final draft.
- Rose Thorogood conceived and designed the experiments, analysed the data, contributed reagents/materials/analysis tools, authored or reviewed drafts of the paper, approved the final draft.

### Animal Ethics

The following information was supplied relating to ethical approvals (i.e. approving body and any reference numbers):

Wild birds were used with permission from the Central Finland Centre for Economic Development, Transport and Environment and licence from the National Animal Experiment Board (ESAVI/9114/ 04.10.07/2014) and the Central Finland Regional Environmental Centre (VARELY/294/2015).

### Data Availability

The raw data is available in the Supplemental Files.

### Supplemental Information

Supplemental information for this article can be found online at http://dx.doi.org/10.7717/ peerj.7998#supplemental-information.

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
