# Peer review of "The effect of social information from live demonstrators compared to video playback on blue tit foraging decisions"

_PeerJ, doi:10.7717/peerj.7998_

## Round 0.1 · original submission · Major Revisions

Please address all of the points made by the reviewers, and please respond in full to the concerns expressed by Reviewer 2.

Reviewer 1 ·

Basic reporting

The text is clearly written, in an unambiguous and professional English. The results are relevant to the hypotheses. The paper is relevant for scholars designing experiments with blue tits (and other similar models), in that it reports the difficulties of obtaining social learning of foraging choices in this species. I think that this paper could also stimulate future attempts to tackle this issue, taking advantage of the results reported here, by developing alternative methodologies on the basis of this experience. For example, it is clearly of great importance to carefully select the landmarks associated with the feeding positions in order to avoid the development of idiosyncratic preferences. Moreover, the information provided by this work helps to clarify the interpretation of previous results, which showed that video playbacks failed to influence foraging decisions. Based on the result of the current paper, it seems unlikely that this was due to the use of video playbacks, suggesting that social learning about food sources might be absent or at least difficult to demonstrate in this species.
The introduction gives sufficient background for the studies, and the relevant literature is adequately represented in the references. However, it could be interesting to mention also the literature on birds’ perception of video animations, in addition to the studies using video playbacks. In fact, there are a number of studies showing that at least young domestic chicks can attribute a social valence to simple animations showing point light displays of biological motion, or even 2D shapes characterised by animate motion features.
Raw data have been shared as supplementary materials. The article is well structured, with clear and informative figures.

Experimental design

This paper represents a piece of original primary research, based on a research question that is well defined, relevant and meaningful in the light of the field literature. Given the importance of video playback experiments for research on social cognition and social learning, it is important to assess how the use of video stimuli affects social learning in this model species. The use of a control group exposed to live conspecifics makes the research more methodologically sound, allowing to exclude possible confounds and alternative explanations. This is particularly important in the present case, since the results did not reflect the initial expectations and could be otherwise particularly difficult to interpret.
This is overall a rigorous investigation performed to the highest standard, whose methods are for most parts described to a sufficient detail to allow replication, with few exceptions (see below).

Validity of the findings

Underlying data have been satisfactorily provided as supplementary materials. Overall the methods and the analysis are statistically sound and well controlled. The conclusions are clear and are directly supported by the results.

Additional comments

Overall there is not much to comment or correct, in the present paper. As testified also in the supplementary information provided, this manuscript has already been the object of a very rigorous and detailed reviewing, bringing it to a very high standard. Most of the weaknesses have already been addressed, corrected or discussed. I have only a few comments that I report here below, in order of importance.

I was wondering if the authors could record also the number of total choices emitted for each well, in addition to the first choice. This is relevant because, even after the lid covering each well has been removed, the “correct” well could still be identified based on its position and the symbol presented in the front of the tray. Total choices are sometimes more sensitive a measure than first choice, and could reveal underlying social learning effects not detectable based on the first choice only.

I find the title to be potentially misleading. I recognise that it states the original question that motivated the paper, but it does not reflect the main finding the results actually provided (blue tits’ low tendency to rely on social information for foraging, regardless of the format of the presentation, live or through video). I suggest that the Authors could rephrase the title to better reflect that?

I would suggest to add some information on the procedure used to restrict access to the food at lines 152-153 (some information about this is provided at lines 217-219). Specifically, could the authors provide some estimation of the level of food motivation caused by the 1h deprivation in this species? Is this a severe or moderate level of deprivation, considering the metabolic needs of this species? The amount of food deprivation can affect the animals’ motivation and attention to social information; in my experience, if an animal is too food motivated it will not pay attention to social information, being too focused on food itself.

Could the authors provide the videos used as stimuli (e.g. in the supplementary materials)?

·

Basic reporting

Video playback provides a powerful experimental approach to manipulate social information in many learning paradigms. As this paper shows, this method is especially relevant when the responses to video stimuli by observers are compared with live presentations, using the same demonstrators. Overall, the paper is well organized and clearly written, using professional language. Figure 1 is excellent; Figure 2 needs some changes to improve clarity (see my comments below and annotations on the pdf). Thanks are due to the authors for providing their raw data. Below, I suggest several references not taken into account by the authors in their introduction.
87-89: Ware et al. (2015) address all these issues and innovate in presenting a double teleprompter technique to simulate live social interaction in pigeons. Importantly, they demonstrate that image presentation rate (frames/s) can be critical in determining a social response behavior. In birds, a 60 Hz refresh rate (at the very least) should be used for editing the video clip. Some exceptional avian displays may require an even higher refresh rate to be perceived as smooth and “natural” by observers. These aspects are not considered in the introduction and methods section.
83: Carouso-Peck and Goldstein (2019) brilliantly show that song copying by young male zebra finches improves significantly when social reinforcement – a video playback of a female exhibiting fluff-up behavior – is presented contingently with the auditory stimulus. A 60 Hz refresh rate was used appropriately for editing the video stimulus.
89-91: Thin film transistor liquid crystal display (TFT-LCD) monitors are flickerless and became available on the market in the early-90s (see Kuo 2013). Ikebuchi & Okanoya (1999) first demonstrated the superiority of the TFT-LCD monitor over the CRT monitor in triggering male courtship song, both in zebra finches and Bengalese finches. Importantly, the same female birds were presented live for comparison with video playback.
References:
Carouso-Peck, S. & Goldstein M.H. 2019. Female social feedback reveals non-imitative mechanisms of vocal learning in zebra finches. Curr. Biol. 29: 1-6.
Ikebuchi, M. & Okanoya, K. 1999. Male zebra finches and Bengalese finches emit directed songs to the video images of conspecific females projected onto a TFT display. Zool. Sci. 16: 63–70 (1999).
Kuo, Y. 2013. Thin film transistor technology - past, present, and future. The Electrochemical Society Interface, Spring issue: 55-61.
Ware, E, Saunders, D.R. & Troje, N.F. 2015. The influence of motion quality on responses towards video playback stimuli. Biol. Open, 4: 803-811.
Figures:
Fig. 2a: percentage values have been calculated separately for each condition (control, video and live). Results for each condition are compared to the expected probability (dotted line). This is fine except that the y-axis title should explicitly say: percentage of tested individuals ("per condition").
Fig. 2b plots the “percentage of tested individuals” that matched or did not match the demonstrator’s choice. In this case, the true percentage values are 15+20+35+30 = 100%.
Please, note that this information has been added on the Figures only to illustrate my point (see pdf).

Experimental design

Only one demonstrator (#134) was filmed retrieving food from a well, which was marked with a cross in one case or with a square in another case. Why not using the same design for all five demonstrators?
193: the sentence should read: …whereas the well covered with plain white was always empty.
203: Following up the point made by Ware et al. (2015), which recording mode was used when filming the demonstrators: progressive or interlaced? And what was the refresh rate (frames/s)?
207: How many control videos were edited?
255: the size of the demonstrator on the screen was smaller (70%) than the size of the live bird. Please, remove “slightly”.
289: Negative binomial are used for count data. Time to event measurements always have positive values and are truncated, which is not the case for a normal or binomial distribution. For latency data, Jahn-Eimermacher et al. (2011) recommend Cox regression analysis.
Jahn-Eimermachera, A., I. Lasarzikb & J. Raber (2011) Statistical analysis of latency outcomes in behavioral experiments. Behav Brain Res. 221: 271-275.

Validity of the findings

Overall, it is clear that symbols acted as distracting stimuli for the blue tits. The presentation of symbols also increased the variation among presentations instead of reducing it, which contradicts a good point made by the authors in the introduction (l. 79). As the authors report (l. 313), Cox regression analysis shows that birds started the task faster after seeing the video. In addition, it shows a shorter latency for the right-side of the tray.
coxph(formula = Surv(lat) ~ choiceloc + info, data = d)
n= 39, number of events= 39
(1 observation deleted due to missingness)
coef exp(coef) se(coef) z Pr(>|z|)
choicelocmiddle 0.2592 1.2959 0.4820 0.538 0.59079
choicelocright 0.9020 2.4646 0.4321 2.088 0.03683 *
infovideo 1.1575 3.1821 0.4072 2.843 0.00447 **

Likelihood ratio test= 9.26 on 3 df, p=0.02607
Wald test = 8.78 on 3 df, p=0.03235
Score (logrank) test = 9.01 on 3 df, p=0.02913

Looking at socially-educated birds only:
(1) Eight birds out of eight (100%) failed to “match” their choice using information associated with two of the five demonstrators (#71 and #105). Not surprisingly, both demonstrators were filmed retrieving food from a well marked with a cross (non-preferred symbol).
(2) By contrast, 5/8 birds (62.5%) did “match” their choice using information associated with two other demonstrators (#92 and #99). Not surprisingly, both demonstrators were filmed retrieving food from a well marked with a square (preferred symbol).
(3) Finally, 2/4 birds (50%) successfully matched their choice based on social information acquired from one demonstrator (#134), which was filmed retrieving food from a well marked with a square (preferred symbol). The two other birds failed to match their choice when the same demonstrator was shown retrieving food from a well marked with a cross (non-preferred symbol).
These results can be tabulated as follows:
demonstrator 71-105-134 92-99-134
symbol cross square total
match 0 7 7
no match 10 3 13
total 10 10 20

Fisher exact test: P = 0.002

Importantly, the seven birds that did match a demonstrator’s choice acquired information - social and/or symbolic - roughly equally from live (3/7; 42.9%) and video (4/7; 57.1%) demonstrators (Fisher exact test: P = 0.562). A Cox regression analysis shows that the latency was significantly shorter for birds that did match their foraging location on the demonstrator's choice, irrespective of social context (live: 60.7 s vs.979.6 s; video: 95.8 s vs. 521.7 s).
coxph(formula = Surv(lat) ~ match, data = data)
n= 20, number of events= 20
coef exp(coef) se(coef) z Pr(>|z|)
matchyes 1.058 2.879 0.539 1.962 0.0498 *

Additional comments

Owing to the presence of a confounding variable (symbol preference), the study fails to reach its goal in clearly demonstrating whether or not blue tits use “social” information provided by video recordings vs. live demonstrators. The authors acknowledge their inconclusive results on several occasions in their discussion (highlighted text on lines 330, 340, 366, 404).
The use of symbols during training obviously distracted the birds from acquiring social information based on foraging location by demonstrators. Similarly, the use of symbols at testing distracted the birds from making a choice purely based on social information acquired during training.
I understand the effort put in those experiments and that the perspective of trapping another group of unmarked blue tits will be time consuming. Unfortunately, if we are to make progress in understanding how birds use social information from video vs. live demonstrators, I can only recommend testing another group of blue tits in the absence of distracting symbols.
The strong preference of blue tits for square symbols, which is not seen in great tits, is really interesting and begs for an explanation. Are the brains of blue tits and great tits wired differently? What would be the ecological significance for blue tits of having visual neurons selective for square outlines? These issues are of course outside the scope of this study.

External reviews were received for this submission. These reviews were used by the Editor when they made their decision, and can be downloaded below.

---

## Round 0.2 · Minor Revisions

Please make the corrections suggested by Reviewer 2.

Reviewer 1 ·

Basic reporting

In this revised version of the manuscript, the authors have adequately addressed all the points raised in the reviewers' reports.

Experimental design

See above.

Validity of the findings

See above.

Additional comments

See above.

·

Basic reporting

'no comment'

Experimental design

'no comment'

Validity of the findings

'no comment'

Additional comments

The authors have responded to all my previous concerns. The new title of the paper is consistent with the revised version. It is fully adequate and I am glad the authors took this initiative. The revised analysis of the data demonstrates that, to some degree, social information acquired from live and video presentations does affect the foraging decisions of blue tits. Both figures are now excellent. The video is superb and very helpful. The experiments were conducted with great care and the paper will make a useful contribution to this exciting field of research. A few minor changes may improve the text before publication:

l. 57 … and even insects… even is not necessary.

l. 90 Here, I feel that my early work with zebra finches - using a CRT video monitor - should be cited: Adret 1997 Discrimination of video images by zebra finches (Taeniopygia guttata): Direct evidence from song performance. J. Comp. Psychol. 111: 115-125.

l. 223 the number of demonstrations for each symbol was balanced…

l. 232 see an examplar video clip…

l. 294 influenced the birds’ latency

l. 295 we also recorded the elapsed time (s)

l. 380 did not infuence the birds’ choices

l. 399 than birds that chose the left side of tray

l. 401: but this effect was marginal (effect of symbol (square): coefficient = 0.918 ± 0.526, Z = 1.745, p = 0.08)

l. 427: they responded similarly to video and live presentations.

l. 436: … juvenile females being more likely to learn from

l. 444 …suggesting that acquired social information did influence their behaviour. (as emphasized on l. 464).

l. 470: …were also found to prefer squares in another experiment

l. 472: A conspicuous square symbol…

External reviews were received for this submission. These reviews were used by the Editor when they made their decision, and can be downloaded below.

---

## Round 0.3 · accepted · Accept

Your paper will make a significant contribution to the field.

External reviews were received for this submission. These reviews were used by the Editor when they made their decision, and can be downloaded below.